# Using dynamic microsimulation to project cognitive function in the elderly population

**Yifan Wei**○*, **Hanke Heun-Johnson**○, **Bryan Tysinger**○

Leonard D. Schaeffer Center for Health Policy and Economics, University of Southern California, Los Angeles, California, United States of America

* weiyifan@usc.edu

## Abstract

### Background

A long-term projection model based on nationally representative data and tracking disease progression across Alzheimer's disease continuum is important for economics evaluation of Alzheimer's disease and other dementias (ADOD) therapy.

### Methods

The Health and Retirement Study (HRS) includes an adapted version of the Telephone Interview for Cognitive Status (TICS27) to evaluate respondents' cognitive function. We developed an ordered probit transition model to predict future TICS27 score. This transition model is utilized in the Future Elderly Model (FEM), a dynamic microsimulation model of health and health-related economic outcomes for the US population. We validated the FEM TICS27 model using a five-fold cross validation approach, by comparing 10-year (2006–2016) simulated outcomes against observed HRS data.

### Results

In aggregate, the distribution of TICS27 scores after ten years of FEM simulation matches the HRS. FEM's assignment of cognitive/mortality status also matches those observed in HRS on the population level. At the individual level, the area under the receiver operating characteristic (AUROC) curve is 0.904 for prediction of dementia or dead with dementia in 10 years, the AUROC for predicting significant cognitive decline in two years for mild cognitive impairment patients is 0.722.

### Conclusions

The FEM TICS27 model demonstrates its predictive accuracy for both two- and ten-year cognitive outcomes. Our cognition projection model is unique in its validation with an unbiased approach, resulting in a high-quality platform for assessing the burden of cognitive decline and translating the benefit of innovative therapies into long-term value to society.

**Data Availability Statement:** Please access the simulation code from this public repository: https://schweb.med.usc.edu/svn/tics27_validation. The data required for this study are available through the Health and Retirement Study. HRS is owned by

The University of Michigan Survey Research Center. To request data access, please access the HRS sharing site https://hrsdata.isr.umich.edu/data-products/contributed-projects or contact hrsquestions@umich.edu.

**Funding:** This research was supported by National Institute on Aging under grants R01AG062277 and P30AG024968. HHJ was supported under R01AG062277. YW was supported under P30AG024968. BT received and was supported under P30AG024968.

**Competing interests:** NO authors have competing interests

## Introduction

Alzheimer's disease and other dementias (ADOD) impose an increasing burden on the United States society and health care system. According to the Alzheimer's Association, the number of Americans 65 and older living with Alzheimer's dementia is estimated to grow from 5.8 million in 2019 to 13.8 million by 2050, as the baby boom generation ages [1]. On the other hand, the US Food and Drug Administration recently approved Aduhelm (Aducanumab), the first disease-modifying treatment (DMT) for ADOD, with more potential DMTs in the pipeline [2,3]. With the approval of Aduhelm, discussion arises about the therapies' real-world value and their potential costs to the healthcare system. As treatments shift in focus from dementia to earlier disease phases like mild cognitive impairment (MCI), concerns are rising about the high upfront costs for initial screening and diagnostics together with late-occurring and uncertain benefits [4,5].

Facing great opportunities and challenges in ADOD therapy development, it is important that we have the proper analysis tools to evaluate the economic impact of cognitive impairment and potential therapies. A cognitive model for use in projecting the US population should model all stages of cognitive decline for a nationally representative sample. Additionally, incorporating predictors and risk factors will make the model useful for assessing counterfactual scenarios and interventions. We identified six ADOD economic evaluation models for the US in the literature, and none of the existing models were based on data nationally representative of the US population [6–11]. Four models used the Uniform Data Set from the US National Alzheimer's Coordinating Center [6,8,10,11]. The Uniform Data Set contains data from the Alzheimer disease centers across the United States, but it is not considered a population-based sample since the enrollment of patients by participating Alzheimer disease centers is not random [6]. One model was based on the Alzheimer's Disease Neuroimaging Initiative (ADNI) database, which is a research cohort of participants in cognitively normal, MCI and dementia states and also not commonly regarded as typical of the current clinical ADOD population [7,10]. One model was not based on a selected sample of participants but used model input parameters from multiple data sources [9]. Among the six models that used US data, three models tracked individuals across the full Alzheimer's disease continuum [7,9,11]. We also identified three models for other countries that tracked individuals across the full Alzheimer's disease continuum [12–14]. Among them, one model was based on six longitudinal cohort studies from different countries; one model for the UK and one model for Spain used model input parameters from multiple data sources. The Spain model represented the Spanish population aged 40 years or older from 2010 to 2050 and was validated against published life expectancy and incidence and prevalence of the dementia stages in Spain [13].

Among the ADOD economic evaluation models we identified, only one model reported rigorous validation. The Alzheimer's Disease Archimedes Condition-Event Simulator was validated by comparison of risk of mortality, institutionalization, and transition to Alzheimer's dementia predictions to external data from patient registries, clinical trials and literature. In each validation, the simulation cohort's baseline characteristics were matched to the study population in external data [7]. Several other models reported comparison of estimated transition probabilities, cognitive trajectories, incidence of dementia and survival to published literature [6,8–11,13]. However, none of the ADOD economic evaluation models we identified in the literature validated their models by comparing simulated output against population-based samples. Model validation is a general challenge in the area of ADOD and many models suffer from this limitation, due to the lack of publicly available data [10].

In this paper, we introduce and validate a microsimulation model to project trajectories in cognitive test scores, FEM TICS27, across the full Alzheimer's disease continuum, based on

nationally representative data of the US population aged 51 and older from the Health and Retirement Study (HRS). We compare our model's 10-year predictive performance against longitudinal HRS data, using area under the receiver operating characteristics curve (AUROC) [15] and five-fold cross validation. We also introduce our validation framework which we believe is valuable to future microsimulation validation studies.

## Methods

### a. The Future Elderly Model (FEM) Overview

The Future Elderly Model (FEM) is a microsimulation model of health and economic outcomes for the US population aged 51 and older. Here we summarize FEM's core functions; technical details are described in a technical appendix [16]. FEM uses first-order Markov transition models to simulate individuals' aging progress. It captures outcomes including health conditions, functional status, earnings and employment status, participation in government benefit programs, and mortality. FEM has been used in many studies answering important policy questions in aging and dementia [17,18]. In this study, we developed and validated a new model for cognition measurement, TICS27 score, in FEM.

### b. Data and measures

FEM uses data from the HRS, a biennial nationally-representative longitudinal survey in the population with more than 37,000 respondents over age 50 in the US. Baseline interviews with existing birth cohorts have been conducted in 1992, 1993, 1998, 2004, 2010, and 2016 with oversampling of Hispanics and African-Americans. Every six years, the HRS enrolls a new birth cohort in order to maintain a steadystate of the US population over age 50. Participants are followed through the life course with the core biennial surveys and supplemental data collections. Technical details on HRS sampling design, recruitment, and measurement are published before [19]. In this validation study, our simulation sample consisted of HRS respondents age 53 or older in 2006 in the 2006 HRS survey. All population-level analyses used HRS sample weights.

With its goal of understanding the challenges and opportunities of aging, HRS includes a section on cognition, since decline in cognitive functioning is a hallmark of aging and predictive of mortality [20]. HRS uses two different sets of measures to assess cognitive status: for respondents who complete the survey themselves ("self-respondents"), cognitive functioning is assessed using an adapted version of the Telephone Interview for Cognitive Status (TICS) [20,21]. For respondents who are not able to complete the survey themselves, questions about changes in memory in the last two years are asked to proxy respondents in the HRS.

TICS is modeled after the Mini-Mental State Exam (MMSE) for use over the telephone, and TICS scores can be converted to MMSE scores using a validated crosswalk [20,22]. TICS tests respondents' cognitive impairment and dementia status, and contains test items that evaluate memory, concentration and executive function, for example by immediate and delayed word recall, counting back from 100 by 7's, and counting back from 20. Composite scores using these test items create a measure of cognitive functioning ranging from 0 to 27 (TICS27) [20,23]. Respondents with scores from 0 to 6 are classified as having dementia, from 7 to 11 as having MCI, and from 12 to 27 as being cognitively normal. This approach was developed and validated by Langa and Weir (2010) [24] and has since been used by many studies on cognitive functioning [23,25,26]. To reduce measurement error in categorizing cognitive status based on TICS27, we require two consecutive responses for dementia: one wave with dementia followed by either dementia or death in the next wave. For MCI, we require either one wave with MCI followed by MCI, dementia or death, or one wave with dementia followed by MCI. All other

cases are categorized as cognitively normal. Cognitive status is considered an absorbing state; once a respondent has been classified with "verified" dementia or MCI, we assume their cognitive status cannot revert to a less severe state.

Our model's target, TICS27 score, is missing from all respondents using a proxy respondent in HRS. Some respondents cannot participate in the interview because of cognitive problems; others might choose to use a proxy because they were working and were thus more likely to be cognitively normal. For example, in 2016, among the 941 proxy respondents, 41.9% did not think the respondent had any cognitive limitations, 6.2% thought the respondent may have some cognitive limitations and 52.0% thought the respondent had cognitive limitations that prevented him or her from being interviewed. For detailed proxy interview cognitive impairment ratings from 2006–2016 HRS, please see Technical Table A-1 in S1 Appendix. This missingness of TICS27 among respondents using a proxy was therefore assumed to be correlated with respondents' cognitive functioning, depending on the reason for using a proxy. Since this missingness is not at random, and HRS does not provide imputed TICS27 values for respondents with a proxy, we adopted a multiple imputation strategy based on HRS' approach for missing TICS27 among self-respondents. We used a combination of relevant demographic, health, and economic variables, as well as prior wave cognitive variables to perform the imputation using the sequential regression method [27]. The multiple imputation was performed using the multiple imputation (mi) command in Stata Version 16.0. Following HRS's practice, we did not impute for participants who were non-responsive to the survey in a given wave.

## c. Key transition models

FEM transition models are a mixture of continuous, binary, and categorical outcomes, with a timescale that mimics the two-year structure of the HRS data. The marginal effects of two specific transition models from the FEM that are most relevant to this study, the TICS27 and mortality transition models, are shown in Table 1. For model coefficients of these two transition models, please refer to Technical Table A-2 in S1 Appendix. The TICS27 transition model was estimated with an ordered probit model using HRS data from 2008 to 2016. An ordered probit model was chosen because TICS27 is a ranked score ranging from 0 to 27. As can be seen in Table 1, variables in the TICS27 model are the TICS27 score from two and four years prior, cognitive status, demographic variables (age, gender, race and ethnicity, education), chronic disease indicators, employment, widowhood, smoking status and body mass index. Our choice of variables aligns with evidence on risk factors for cognitive decline in existing literature [28]. Also, we limited variables to information that is readily available from the survey, i.e. not based on blood samples, genetic data, or other clinical procedures. The mortality transition model was estimated with a probit model using HRS data from 2008 to 2016. Prior validation shows that FEM projections on mortality are generally in line with observed mortality rates [29].

## d. Model validation approach

We validated the FEM TICS27 model at both the population- and individual-levels. At the population-level, we looked at two outcomes, TICS27 distribution comparisons and 10-year changes in a composite measure of both cognition and mortality. At the individual-level, we assessed FEM's performance in predicting dementia in 2/4/6/8/10 years and significant decline in TICS27 in two years using receiver operating characteristics (ROC) curves.

We validated the FEM's TICS27 model using a five-fold cross validation approach, by comparing 10-year simulated population- and individual-level outcomes against observed HRS data in 2016. All validation analyses results are based on 500 Monte Carlo simulations of FEM.

**Table 1. Marginal effect from FEM TICS27 and mortality transition model.**

| Variables | TICS27 Coefficient (Std. Err) | Mortality Coefficient (Std.Err) |
|---|---|---|
| Two-year lag TICS score | 0.0915 (0.0015)*** | |
| Four-year lag TICS score | 0.0955 (0.0017)*** | |
| Non-Hispanic Black | -0.2343 (0.0116)*** | 0.0028 (0.0023) |
| Hispanic | -0.1502 (0.0144)*** | -0.0094 (0.0031)** |
| Did not graduate high school | -0.1361 (0.0131)*** | -0.0002 (0.0022) |
| At least some college | 0.1967 (0.0092)*** | -0.0072 (0.0019)*** |
| Male | -0.0884 (0.0086)*** | 0.0177 (0.0019)*** |
| Slope of age spline before age 65 | -0.0009 (0.0013) | 0.0025 (0.0004)*** |
| Slope of age spline ages 65–74 | -0.0173 (0.0015)*** | 0.0029 (0.0003)*** |
| Slope of age spline ages 75 and older | -0.0353 (0.0013)*** | |
| Slope of age spline ages 75–84 | | 0.0040 (0.0003)*** |
| Slope of age spline ages 85 and older | | 0.0067 (0.0004)*** |
| Ever diagnosed with heart problems | -0.0126 (0.0108) | 0.0131 (0.0020)*** |
| Ever diagnosed with stroke | -0.1288 (0.0165)*** | 0.0110 (0.0023)*** |
| Ever diagnosed with cancer | 0.0008 (0.0117) | 0.0334 (0.0020)*** |
| Ever diagnosed with hypertension | -0.0355 (0.0088)*** | 0.0097 (0.0019)*** |
| Ever diagnosed with diabetes | -0.0693 (0.0105)*** | 0.0155 (0.0020)*** |
| Ever diagnosed with lung disease | -0.0430 (0.0147)** | 0.0294 (0.0023)*** |
| Heart attack in past 2 years | -0.0750 (0.0334)* | 0.0049 (0.0047) |
| Working for pay | 0.0943 (0.0100)*** | |
| Widowed | -0.0298 (0.0121)* | 0.0055 (0.0022)* |
| Ever smoked | -0.0426 (0.0083)*** | |
| Verified ADOD/MCI ever | -0.7105 (0.0155)*** | |
| Delta age | -0.1374 (0.0208)*** | |
| Lag log BMI below 30 | 0.1314 (0.0350)*** | |
| Lag log BMI above 30 | -0.0090 (0.0441) | |
| Difficulty with one IADL | | 0.0162 (0.0028)*** |
| Difficulty with two or more IADLs | | 0.0483 (0.0031)*** |
| Difficulty with one ADL | | 0.0222 (0.0025)*** |
| Difficulty with two ADLSs | | 0.0342 (0.0033)*** |
| Difficulty with three or more ADLs | | 0.0603 (0.0028)*** |
| Current smoker | | 0.0187 (0.0027)*** |
| Diagnosed with heart problems by age 50 | | 0.0051 (0.0067) |
| Diagnosed with stroke by age 50 | | -0.0107 (0.0167) |
| Diagnosed with cancer by age 50 | | -0.0083 (0.0057) |
| Diagnosed with hypertension by age 50 | | 0.0027 (0.0042) |
| Diagnosed with diabetes by age 50 | | 0.0142 (0.0034)*** |
| Diagnosed with lung disease by age 50 | | -0.0278 (0.148) |
| Ever smoked at age 50 | | 0.0061 (0.0022)** |
| Current smoker at age 50 | | 0.0174 (0.0024)*** |
| Ever diagnosed with congestive heart failure | | 0.0259 (0.0028)*** |

Notes: *, significant at $\alpha = 0.05$

**, significant at $\alpha = 0.01$

***, significant at $\alpha = 0.001$.

A five-fold cross validation approach allows us to have separate datasets for estimation and simulation to evaluate FEM TICS27 model's performance in an independent dataset. Cross-validation is a data resampling method to assess the generalizability of predictive models and to prevent overfitting [30]. To do this, we first randomly partitioned our simulation sample into five complementary subsets. We then saved one subset for simulation and used the other four subsets to estimate transition models for this simulation. We repeated this process five times so that each subset was used once for simulation. Finally, we pooled results from five simulations on subsets together for validation analyses. We chose to use five folds since it has been shown empirically that a five- to ten-fold cross validation is the optimal approach [31].

Two different samples were used in the validation analyses. One was the complete 2006 HRS sample, which included HRS respondents age 53 or older in 2006. This sample was used in population-level distribution comparison analyses. The other was the 2006 HRS with full 10-year follow-up sample, which was a subset of the complete 2006 HRS sample and used to determine the population-level 10-year change in cognitive/mortality status, and individual-level analyses. This full 10-year follow-up sample required individuals to respond to every wave of the HRS survey from 2006 to either 2016 or their death.

Population-level outcomes include TICS27 distribution comparisons and 10-year changes in a composite measure of both cognition and mortality. We adopted this composite measure since people with dementia have high mortality rates. Individual-level outcomes include predicting dementia status in 10 years and predicting decline larger than 3 TICS27 points within 2 years for patients with MCI.

On a population-level, the distribution of simulated TICS27 in 2016 was compared to the 2016 HRS population in the same age range (age 63 or older). We also analyzed the 10-year change in status by comparing assignment of cognitive status or death by FEM in 2016 given the 2006 cognitive status to the observed status in HRS. Cognitive status at death was determined by the cognitive status in the last wave before death.

On the individual level, we assessed FEM's performance in predicting dementia in 10 years and significant decline in TICS27 in two years using receiver operating characteristics (ROC) curves. Though more commonly used in regression-based risk prediction models, ROC curves have been used for validation of other disease simulation models as well [15,32]. For individual-level analyses, we ran FEM 500 times over a 10-year time horizon for every individual in the 2006 HRS full 10-year follow-up sample. After 500 simulation iterations, we calculated the percentages of iterations for every individual with specific outcomes for two measures: (1) alive or dead with dementia in 2016; (2) significant decline (decline greater than or equal to 3 points) in TICS27 in 2008. Prior research found that a 3-point decline in MMSE indicated significant decline [33,34]. Using a crosswalk between MMSE and TICS27, a 3-point decline in MMSE translates to a 3-point decline in TICS27 for people with MCI (MMSE from 24 to 27) [35]. We then ranked every individual in the simulation by their FEM-based risks for each separate measure. These ranks were compared to their actual outcome in the HRS data to generate ROC curves. We used area under the ROC curve (AUROC), which is a commonly used measure for predictive model performance, to evaluate FEM's performance on these two measures.

We also compared our model's performance to one of the best-performing models for predicting cognitive decline, COMPASS [36]. COMPASS used data from the ADNI database with information on age, gender, education, APOE genotype and cognitive composite scores on memory and executive functions to predict changes in MMSE scores over 24-months. COMPASS evaluated its performance on predicting significant decline in MMSE scores (3 points) in MCI subjects in 2 years using AUROC. We compared FEM's performance on predicting significant TICS27 decline in MCI subjects in 2 years to COMPASS.

The University of Southern California IRB approved this research under UP-18-00776 ("Ensuing Access to Novel Alzheimer's and Dementia Treatments") on November 21, 2019. This is a retrospective study of secondary data from the Health and Retirement Study that is de-identified and publicly available. This study uses HRS Public Release data which is fully anonymized before researchers' access. Prior to each interview, HRS participants are provided with a written informed consent information document. At the start of each interview, all HRS participants are read a confidentiality statement and give oral consent by agreeing to do the interview. Their oral consents are documented in answers to the survey questionnaire. FEM is programmed in C++, SAS and Stata, and all validation analyses were performed using Stata Version 16.0.

## Results

### a. Sample characteristics

Weighted baseline characteristics for the 2006 HRS and the 2006 HRS with full 10-year follow-up samples are shown in Table 2. Since the full follow-up sample includes relatively more

**Table 2. Characteristics of the 2006 Health and Retirement Survey (HRS) respondents.**

| | **2006 HRS** | **2006 HRS w/ full 10-year follow-up** | **P>|t|** |
|---|---|---|---|
| **Characteristics** | | Mean | |
| **N** | 15,764 | 13,106 | |
| **Age** | 66.91 | 67.34 | 0.000*** |
| **Race and Ethnicity** | | | |
| Non-Hispanic White | 0.811 | 0.814 | 0.041* |
| Non-Hispanic Black | 0.091 | 0.093 | 0.056 |
| Hispanic | 0.073 | 0.071 | 0.031* |
| **Education** | | | |
| Less than high school | 0.181 | 0.183 | 0.168 |
| **High School** | 0.347 | 0.348 | 0.654 |
| Some college and above | 0.472 | 0.469 | 0.165 |
| **TICS score** | 15.57 | 15.54 | 0.205 |
| **Verified cognitive status** | | | |
| Dementia | 0.012 | 0.013 | 0.203 |
| Mild Cognitive Impairment | 0.089 | 0.090 | 0.203 |
| Normal | 0.900 | 0.898 | 0.203 |
| **Disease status** | | | |
| Heart disease ever | 0.215 | 0.223 | 0.000*** |
| Stroke ever | 0.075 | 0.080 | 0.000*** |
| Cancer ever | 0.131 | 0.139 | 0.000*** |
| Hypertension ever | 0.517 | 0.525 | 0.001*** |
| Diabetes ever | 0.179 | 0.186 | 0.000*** |
| Lung disease ever | 0.083 | 0.089 | 0.000*** |
| Heart attack | 0.017 | 0.016 | 0.493 |
| **Work for pay** | 0.429 | 0.417 | 0.000*** |
| **Widowed** | 0.177 | 0.186 | 0.000*** |
| **Smoking ever** | 0.571 | 0.577 | 0.006** |

Notes: *, significant at α = 0.05

**, significant at α = 0.01

***, significant at α = 0.001.

respondents who died between 2006 and 2016, respondents are older and have more chronic conditions compared to the 2006 HRS sample. Other variables are comparable between the two samples, including baseline TICS27 score and confirmed cognitive status. In 2006, the mean TICS27 score for the HRS sample and the full follow-up samples were 15.57 and 15.54, respectively (P = 0.21). In the 2006 HRS sample, 1.2% of respondents were living with dementia, 8.9% had MCI and 90.0% were cognitively normal; in the full follow-up sample, 1.2% of respondents were living with dementia, 9.0% had MCI and 89.8% were cognitively normal.

### b. Population-level predictions

Fig 1 shows the distribution of TICS27 in 2006 (grey line) and the subsequent decline in this cognitive measure in 2016 from FEM simulations (black line) and HRS observations (dashed line), for HRS respondents ages 53+ and 65+ in 2006. Table 3 shows the mean TICS27 score, 10-year change in mean TICS27 score and TICS27 score at different percentiles for both FEM and HRS, for HRS respondent ages 53+ and 65+ in 2006. In aggregate, the distribution of TICS scores after ten years of FEM simulation matches the 2016 HRS distribution well, both at the mean and at specific points in the distribution, for both age groups. For HRS respondents ages 53+ in 2006, the mean ten-year change in TICS27 is -0.62, compared to -0.64 in FEM simulation; at the 10th, 50th and 90th percentiles of 2016 TICS27 distribution, the TICS27 score in HRS is 9, 15, 21, and the TICS27 score in FEM is 9, 15, 20, respectively. For HRS respondents ages 65+ in 2006, the mean ten-year change in TICS27 is -1.51, compared to -1.67 in FEM simulation; at the 10th, 50th and 90th percentiles of 2016 TICS27 distribution, the TICS27 score in HRS is 6, 12, 18, and the TICS27 score in FEM is 5, 12, 18, respectively.

Table 4 shows the 10-year change in distributions of combined cognitive/mortality status given a respondent's initial status. The 2016 status has five categories: cognitively normal, MCI, dementia, dead without dementia, and dead with dementia. Overall, compared to HRS data, FEM assigns similar percentages of people to each cognitive/mortality category in 2016. Of HRS respondents, 56.7% retained normal cognitive function between 2006 and 2016; FEM assigns 58.5% of respondents to this category. In HRS in 2016, 9.9% of respondents were in the MCI category, and 1.9% of respondents were in the dementia category; the predictions from FEM are 9.0% and 2.5%, respectively. In HRS in 2016, 27.0% of respondents were dead without dementia and 4.5% were dead with dementia; FEM predicts 25.5% and 4.5% of respondents to be in these categories, respectively.

### c. Individual-level predictions

Table 5 shows AUROC results for FEM predicting (1) dementia or death with dementia, and (2) dementia conditional on being alive in 10 years, for both the full follow-up sample and sub-population analyses (e.g. by race and ethnicity). Fig 2 shows the ROC curves for the full follow-up sample (Panels A and B) and individuals with MCI in the 2006 sample (Panels C and D). In the full follow-up sample, the AUROC for dementia or dead with dementia in 10 years is 0.904, the AUROC for dementia conditional on being alive is 0.868. FEM's performance on predicting MCI or worse is comparable to that of predicting dementia (Table 5). Furthermore, FEM's predictive performance is comparable for subgroups of age, race and ethnicity, education and disease status. For people aged 65 years or older in 2006, the AUROC for dementia or dead with dementia is 0.875. For non-Hispanic Black and Hispanic people, the AUROC for dementia or dead with dementia is 0.906 and 0.881, compared to 0.891 for non-Hispanic White people. For people without a high school degree, the AUROC for dementia or dead with dementia is 0.866, compared to 0.856 for people with high school education and 0.880 for people with at least some college education. For people who ever had a stroke before

Panel a. Ages 53+ in 2006

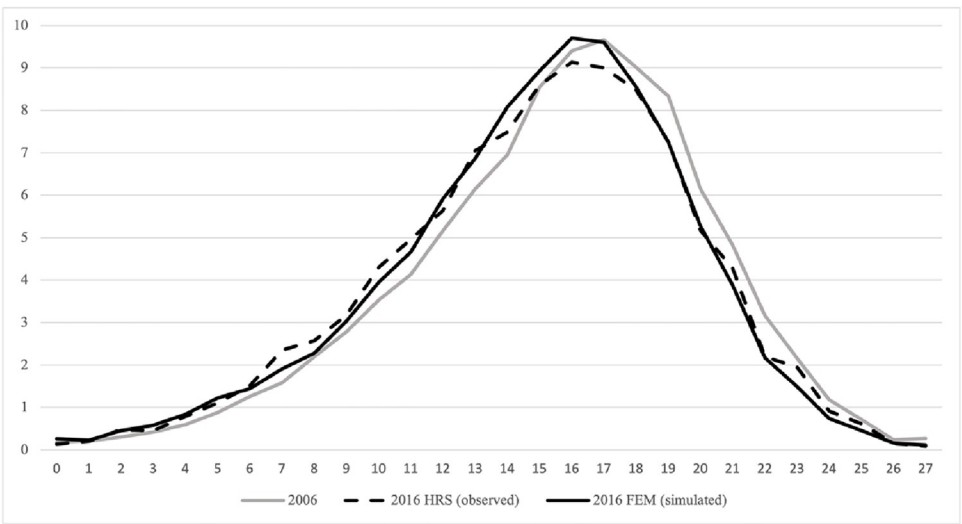

Panel b. Ages 70+ in 2006

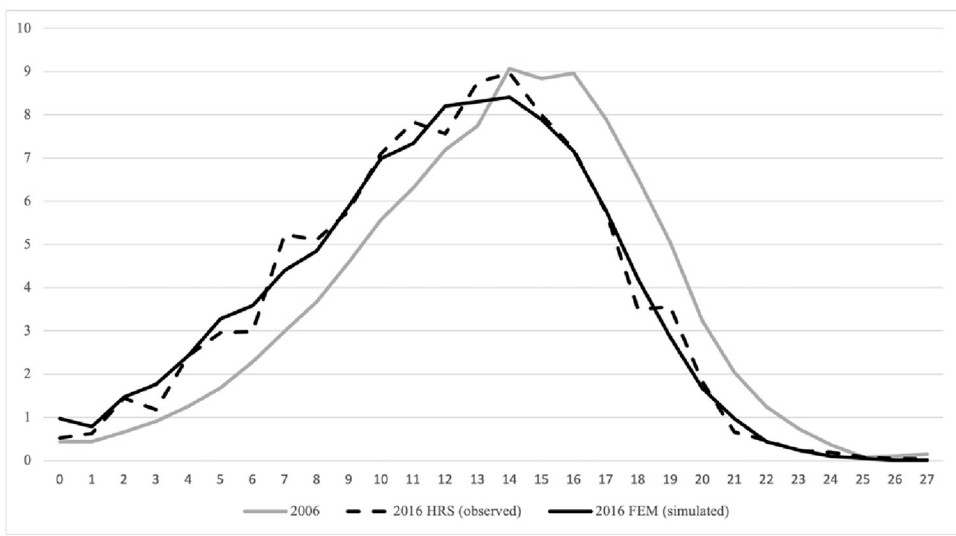

**Fig 1. Distribution comparison between HRS and FEM, 2006–2016.**

2006, the AUROC for dementia or dead with dementia is 0.875. For people with MCI in 2006, the AUROC for dementia or dead with dementia is 0.720, and the AUROC for dementia conditional on being alive is 0.705.

## d. External comparison

Table 7 shows FEM's AUROC results predicting significant decline (greater than or equal to 3 points) in TICS27 in two years and its comparison to the COMPASS model's performance in MCI subjects. The AUROC for FEM on predicting significant decline in TICS27 in two years is 0.722 for people with MCI in 2006. FEM's performance is better than Base COMPASS

**Table 3. Distribution comparison between HRS respondents and FEM simulation, 2006–2016.**

**Ages 53+ in 2006**

|  | 1th | 5th | 10th | 25th | 50th | 75th | 90th | 95th | 99th | Mean | Diff (Mean) |
|---|---|---|---|---|---|---|---|---|---|---|---|
| **2006** | 3 | 7 | 9 | 13 | 16 | 19 | 21 | 22 | 25 | 15.57 | |
| **2016 HRS (observed)** | 3 | 7 | 9 | 12 | 15 | 18 | 21 | 22 | 24 | 14.95 | -0.62 |
| **2016 FEM (predicted)** | 3 | 6 | 9 | 12 | 15 | 18 | 20 | 22 | 24 | 14.93 | -0.64 |

**Ages 65+ in 2006**

|  | 1th | 5th | 10th | 25th | 50th | 75th | 90th | 95th | 99th | Mean | Diff (Mean) |
|---|---|---|---|---|---|---|---|---|---|---|---|
| **2006** | 2 | 5 | 7 | 11 | 14 | 17 | 19 | 20 | 23 | 13.55 | |
| **2016 HRS (observed)** | 1 | 4 | 6 | 9 | 12 | 15 | 18 | 19 | 22 | 12.04 | -1.51 |
| **2016 FEM (predicted)** | 1 | 4 | 5 | 9 | 12 | 15 | 18 | 19 | 21 | 11.88 | -1.67 |

Note: HRS stands for Health and Retirement Study, which is the observed survey data. FEM stands for Future Elderly Model, which generates the simulated results.

(AUROC 0.641), which is a machine learning model that additionally uses APOE genotype information. Advanced COMPASS (AUROC 0.814) outperforms FEM, although Advanced COMPASS includes information not only on APOE genotype but also on neuropsychological tests and validated composite scores for memory and executive functions.

## Discussion

We extended the FEM microsimulation model to include a widely used cognitive test based on nationally representative HRS data, using individual-level information on demographics (age, gender, race and ethnicity, education), chronic disease indicators (heart disease, stroke, cancer, hypertension, diabetes, lung disease, heart attack), employment, smoking status, marital status and body mass index. The FEM TICS27 model can be used to estimate the future burden and long-term value of treatments of cognitive decline in the US. It also provides a 10-year risk score for dementia based on information attainable from a telephone-based survey.

To our knowledge, most disease simulation models for cognitive decline and dementia are not validated or are not validated using an unbiased approach like five-fold cross validation. Given the limited access to data and adoption of different cognitive function tests, validation of modeling methods is a general challenge in the area of ADOD [10]. We are not aware of data sources other than the HRS using TICS27 as cognitive function measurement that are available as independent datasets for external validation. Adoption of five-fold cross validation is an improvement compared to most existing economic evaluation models for ADOD in this

**Table 4. 10-year change in distributions of cognitive status based on TICS score, HRS and FEM.**

|  | Cognitively Normal 2016 (%) | | MCI 2016 (%) | | Dementia 2016 (%) | | Dead w/o dementia 2016 (%) | | Dead w/ dementia 2016 (%) | | Total |
|---|---|---|---|---|---|---|---|---|---|---|---|
|  | HRS | FEM | HRS | FEM | HRS | FEM | HRS | FEM | HRS | FEM | |
| **Cognitively Normal 2006** | 56.7 | 58.5 | 7.4 | 6.4 | 0.8 | 1.0 | 23.0 | 22.5 | 1.8 | 1.4 | 89.8 |
| **MCI 2006** | 0 | 0 | 2.5 | 2.6 | 0.9 | 1.2 | 4.0 | 3.0 | 1.7 | 2.2 | 9.0 |
| **Dementia 2006** | 0 | 0 | 0 | 0 | 0.2 | 0.3 | 0 | 0 | 1.0 | 0.9 | 1.3 |
| **Total** | 56.7 | 58.5 | 9.9 | 9.0 | 1.9 | 2.5 | 27.0 | 25.5 | 4.5 | 4.5 | 100 |

Note: HRS stands for Health and Retirement Study, which is the observed survey data. FEM stands for Future Elderly Model, which generates the simulated result.

**Table 5. Area under the receiver operating characteristics curve (AUROC) for predicting dementia from 5-fold cross-validation.**

| Sample | Sample Size | Outcome | AUROC |
|---|---|---|---|
| **Full sample** | 13,106 | Dementia or dead w/ dementia | 0.904 |
| | | Dementia (conditional on alive) | 0.868 |
| | | MCI or worse or dead w/ MCI or worse | 0.897 |
| | | MCI or worse (conditional on alive) | 0.826 |
| **Verified MCI in 2006** | 1,512 | Dementia or dead w/ dementia | 0.720 |
| | | Dementia (conditional on alive) | 0.705 |
| **Non-Hispanic White** | 9,922 | Dementia or dead w/ dementia | 0.891 |
| | | Dementia (conditional on alive) | 0.838 |
| **Non-Hispanic Black** | 1,824 | Dementia or dead w/ dementia | 0.906 |
| | | Dementia (conditional on alive) | 0.814 |
| **Hispanic** | 1,107 | Dementia or dead w/ dementia | 0.881 |
| | | Dementia (conditional on alive) | 0.827 |
| **Less than high school** | 2,983 | Dementia or dead w/ dementia | 0.866 |
| | | Dementia (conditional on alive) | 0.788 |
| **High School** | 4656 | Dementia or dead w/ dementia | 0.856 |
| | | Dementia (conditional on alive) | 0.821 |
| **Some college or above** | 5,467 | Dementia or dead w/ dementia | 0.880 |
| | | Dementia (conditional on alive) | 0.806 |
| **Age 65+ in 2006** | 6,272 | Dementia or dead w/ dementia | 0.875 |
| | | Dementia (conditional on alive) | 0.828 |
| | | MCI or worse or dead w/ MCI or worse | 0.861 |
| | | MCI or worse (conditional on alive) | 0.766 |
| **Ever had diabetes before 2006** | 2,681 | Dementia or dead w/ dementia | 0.884 |
| | | Dementia (conditional on alive) | 0.825 |
| **Ever had stroke before 2006** | 1,239 | Dementia or dead w/ dementia | 0.875 |
| | | Dementia (conditional on alive) | 0.849 |
| **Ever had hypertension before 2006** | 7,445 | Dementia or dead w/ dementia | 0.906 |
| | | Dementia (conditional on alive) | 0.859 |
| **Ever had heart disease before 2006** | 3,312 | Dementia or dead w/ dementia | 0.895 |
| | | Dementia (conditional on alive) | 0.833 |

To demonstrate FEM TICS27's performance across years, Table 6 shows AUROC for predicting the main outcome, dementia or dead with dementia, in 2, 4, 6, 8 and 10 years. As shown, FEM TICS27's predictive performance is highest for 2-year prediction and decreases as the prediction timeframe increases. The 10-year AUROC for the full sample is still above 0.9.

situation to validate model performance. Using the same data for model estimation and validation can lead to an upward bias in model performance estimates due to overfitting. Although k-fold cross validation is one of the most widely used data resampling methods to estimate the true prediction error of models and to tune model parameters in risk prediction models, it is rarely used in validation for disease simulation models. Cross validation enables us to assess the generalizability of a model without using a new independent dataset, which is critical to obtaining unbiased results for model prediction performance [30,31].

The FEM TICS27 model demonstrates excellent internal validity: the TICS27 distribution and 10-year change in cognitive status generated by FEM simulation closely matches observed HRS data, and the AUROCs are larger than 0.85 for dementia prediction. For prediction of significant decline in MCI patients, FEM's performance is comparable to one of the best-performing models reported in the literature [36].

Panel A. 2006 HRS with full 10-year full follow-up sample.

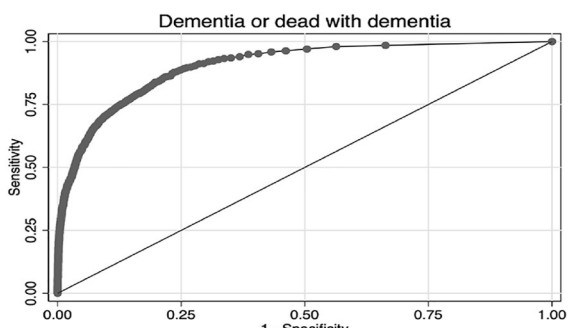

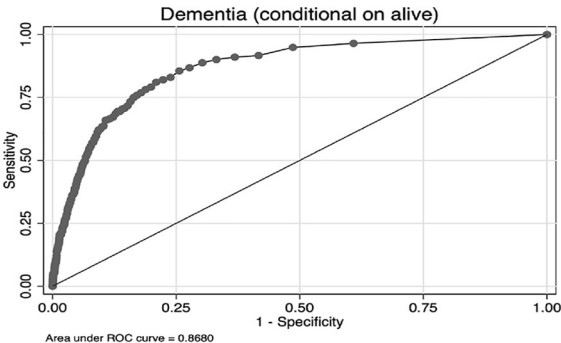

Panel B. MCI in 2006 sample.

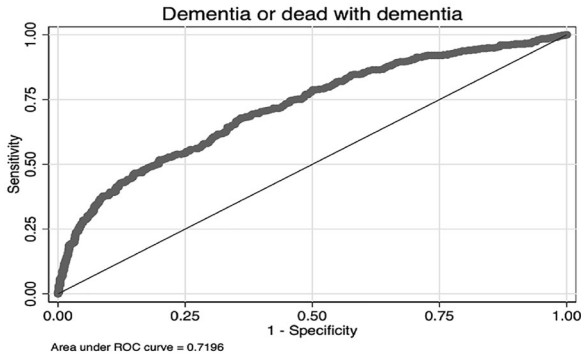

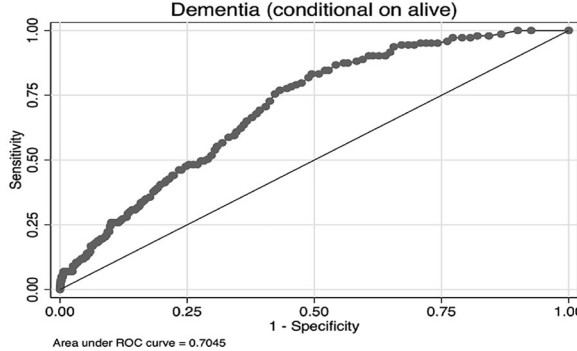

**Fig 2. Receiver operating characteristics curve for predicting dementia in 10 years.**

**Table 6. Area under the receiver operating characteristics curve (AUROC) for predicting dementia or dead with dementia for 2, 4, 6, 8 and 10 years.**

|  | 2 years (2008) | 4 years (2010) | 6 years (2012) | 8 years (2014) | 10 years (2016) |
|---|---|---|---|---|---|
| Full sample | 0.995 | 0.955 | 0.941 | 0.920 | 0.904 |
| Age 65+ in 2006 | 0.993 | 0.946 | 0.929 | 0.902 | 0.883 |
| Verified MCI in 2006 | 0.961 | 0.812 | 0.783 | 0.740 | 0.720 |

FEM TICS27's performance on two individual-level outcomes, long-term prediction of dementia and short-term prediction of cognitive decline, is comparable to or exceeds the performance of existing models. Previously published studies reported AUROCs between 0.6 and 0.78 for prediction of AD/dementia within 3–40 years [37], which is lower than the AUROC of 0.904 reported in this study for prediction of 10-year dementia or dead with dementia. For predictions of significant decline of cognitive test scores in two years, FEM TICS27's performance is comparable with one of the best-performing models, COMPASS, which won the Dialogue for Reverse Engineering And Method (DREAM) Alzheimer's Disease Big Data challenge. One of the drawbacks of COMPASS is that it requires knowledge of detailed clinical information and APOE genotype, and is based on a selective disease registry, the ADNI database [36]. FEM TICS27 on the other hand solely relies on demographic and survey-derived variables, and can provide nationally representative estimates. Thus, FEM TICS27 demonstrated its predictive accuracy for both long-term dementia status and short-term cognitive decline outcomes. The increased performance of FEM over other models is likely because it utilizes information on individual characteristics and behavior, like smoking, widowhood, and disease history.

On the other hand, Advanced COMPASS is better at predicting outcomes for people with MCI, which is especially hard because of heterogeneity in the prognosis and the disease progression with respect to patient characteristics [38]. For this specific group, additional clinical and genotype information significantly improves prediction performance [36]. Future development of FEM TICS27 with genotype, blood-based biomarker variables, behavioral symptoms and history of medication, which are available for a subsample of the HRS, will possibly improve its performance for people with MCI. Additionally, future applications of FEM TICS27 will include analyses of differences in cognitive trajectory by education, initial cognitive status, and race and ethnicity. The model can also be implemented in microsimulations for other countries.

With Aduhelm approved as the first ADOD DMT and more DMTs in the development pipeline, the future looks promising. Though crucial, availability of DMT is only one step in enhancing cognitive function in elderly population. Demonstrating value of treatment and identification of people at risk of cognitive impairment are two very important components as well. FEM microsimulation could help with these. Understanding the long-term impact of ADOD DMTs beyond direct medical expenditure is crucial to its value demonstration [39]. As randomized controlled trials can only generate short-term evidence on the efficacy of ADOD DMTs, to demonstrate their long-term value, projection models are needed to estimate future

**Table 7. Two-year TICS27 significant decline in MCI subjects and comparable results from COMPASS.**

|  | FEM TICS27 MCI | Base COMPASS MCI | Advanced COMPASS MCI |
|---|---|---|---|
| AUROC (random = 0.5) | 0.722 | 0.641 | 0.814 |

Notes: FEM TICS27 is the model developed and validated in this paper. COMPASS is one of the best-performing models reported in the literature, which won the Dialogue for Reverse Engineering And Method (DREAM) Alzheimer's Disease Big Data challenge.

benefits. Based on nationally representative data and modeling a large spectrum of cognitive functioning, FEM TICS27 is a useful tool to assess the long-term impact of these future changes on the US healthcare system. Besides accurately modeling cognitive decline, FEM tracks other relevant outcomes, such as functional limitations, physical health, formal and informal care utilization, nursing home living, and medical care costs. FEM is able to provide much-needed evidence on long-term value of ADOD DMT on a broad range of outcomes. The advantage of FEM TICS27 is its high prediction accuracy using only information from a telephone-based survey.

We present FEM TICS27's model structure, variables, data sources and conduct validation of its simulation outcomes against observed HRS data. We show that FEM TICS27 model can accurately predict cognitive test scores covering the full ADOD disease continuum for a nationally representative sample over a 10-year period. This paper demonstrated FEM TICS27's usefulness as a model for long-term economic evaluation for ADOD.

## Supporting information

**S1 Appendix. Technical appendix.**
(DOCX)

## Acknowledgments

We thank Dr. Jakub Hlávka, Dr. Darius Lakdawalla, Dr. Dana Goldman, Dr. Julie Zissimopoulos and Dr. Duncan Ermini Leaf for their feedback on this work.

## Author Contributions

**Conceptualization:** Bryan Tysinger.

**Formal analysis:** Yifan Wei, Bryan Tysinger.

**Funding acquisition:** Bryan Tysinger.

**Investigation:** Yifan Wei, Hanke Heun-Johnson, Bryan Tysinger.

**Methodology:** Yifan Wei, Hanke Heun-Johnson, Bryan Tysinger.

**Project administration:** Bryan Tysinger.

**Supervision:** Bryan Tysinger.

**Validation:** Yifan Wei, Bryan Tysinger.

**Visualization:** Yifan Wei.

**Writing – original draft:** Yifan Wei.

**Writing – review & editing:** Hanke Heun-Johnson, Bryan Tysinger.

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
