## [Decision Letter · Decision Letter 0]

11 Apr 2022

PONE-D-22-01856Using dynamic microsimulation to project cognitive function in the elderly populationPLOS ONE

Dear Dr. Wei,

Thank you for submitting your manuscript to PLOS ONE. After careful consideration, we feel that it has merit but does not fully meet PLOS ONE’s publication criteria as it currently stands. Therefore, we invite you to submit a revised version of the manuscript that addresses the points raised during the review process.

Kindly enhance the discussion part by elaborating how dynamic microsimulation could enhance the cognitive function in elderly people. Future directions is needed for better research projection.

We look forward to receiving your revised manuscript.

Kind regards,

Lai Kuan Lee

Academic Editor

PLOS ONE

Journal Requirements:

Additional Editor Comments (if provided):

Dear Dr. Wei,

The reviewers are commented on your submission. You are required to revise your manuscript accordingly and submit the revised article in a timely manner.

Reviewers' comments:

Reviewer's Responses to Questions

**Comments to the Author**

1. Is the manuscript technically sound, and do the data support the conclusions?

Reviewer #1: Yes

Reviewer #2: Yes

2. Has the statistical analysis been performed appropriately and rigorously? 

Reviewer #1: Yes

Reviewer #2: Yes

3. Have the authors made all data underlying the findings in their manuscript fully available?

Reviewer #1: Yes

Reviewer #2: Yes

4. Is the manuscript presented in an intelligible fashion and written in standard English?

Reviewer #1: Yes

Reviewer #2: Yes

5. Review Comments to the Author

Reviewer #1: This study is of great significance and has high clinical value.

The FEM TICS27 model demonstrates its predictive accuracy for both two- and ten-year cognitive outcomes. I would like to know what the prediction results of other years are by this model?

Thank you!

Reviewer #2: İn this study present FEM TICS27’s model structure, variables, data sources, and conducts validation of its simulation outcomes against observed HRS data. Information about the method was insufficient in this article. According to the studies in the literature, it has increased performance. However, this study should explain why it differs from other studies.

6. PLOS authors have the option to publish the peer review history of their article (what does this mean?). If published, this will include your full peer review and any attached files.

Reviewer #1: No

Reviewer #2: No

---

## [Author Response · Author response to Decision Letter 0]

17 Aug 2022

We thank the editors and reviewers for their time and attention in improving this manuscript. The comments strengthen the manuscript and add clarity, and we appreciate the opportunity to address these issues. To summarize the changes to the manuscript, we have copied the editor’s and reviewers’ comments below, along with our responses. For detailed responses to reviewers, please refer to the attached 'response to reviewers' file.

Response to Editor

1. Kindly enhance the discussion part by elaborating how dynamic microsimulation could enhance the cognitive function in elderly people. 

We thank the reviewer for this excellent suggestion. We added a section in the Discussion to demonstrate more clearly FEM’s role in enhancing cognitive function in elderly population. We highlighted its usefulness in demonstrating value of treatment and identification of people at risk of cognitive impairment, which are two important components of potentially enhancing cognitive function in the elderly population.

2. Future directions is needed for better research projection.

We would like to thank the reviewer for pointing out this omission. We edited the Discussion section, to include future research directions. We outline two approaches: 1) improving FEM TICS27’s performance by utilizing additional predictors; 2) applying FEM TICS27 to different contexts and countries to analyze differences in cognitive trajectory.

Response to Reviewer 1

1. This study is of great significance and has high clinical value. The FEM TICS27 model demonstrates its predictive accuracy for both two- and ten-year cognitive outcomes. I would like to know what the prediction results of other years are by this model? Thank you!

We appreciate the reviewer’s acknowledgment of our study’s significance. And we agree that it is important to show FEM TICS27’s performance across years. We added Table 6 to present FEM TICS27’s performance for 2, 4, 6, 8 and 10 years in predicting dementia or dead with dementia, and refer to this table in the text.

Response to Reviewer 2

1. In this study present FEM TICS27’s model structure, variables, data sources, and conducts validation of its simulation outcomes against observed HRS data. Information about the method was insufficient in this article. 

We agree with the reviewer that the validation strategy as originally presented in Methods section ‘d. Model validation approach’ may have been confusing. We clarified the approach by adding a high-level overview of the validation strategy in this section. We then continue to explain more technical details of our validation process in the paragraphs that follow. 

2. According to the studies in the literature, it has increased performance. However, this study should explain why it differs from other studies.

We added the following sentence to the Discussion: "The increased performance of FEM over other models is likely because it utilizes information on individual characteristics and behavior, like smoking, widowhood, and disease history".

Unrelated to the editor’s or reviewers’ comments, we found we had incorrectly labeled one of the age categories in our original manuscript. Results were presented for both ages 53+ and 70+, but these should have been labeled with ages 53+ and 65+, respectively. The results remain the same. We have corrected the label in the manuscript and tables.

---

## [Editor Report · Decision Letter 1]

30 Aug 2022

Using dynamic microsimulation to project cognitive function in the elderly population

PONE-D-22-01856R1

Dear Dr. Wei,

We’re pleased to inform you that your manuscript has been judged scientifically suitable for publication and will be formally accepted for publication once it meets all outstanding technical requirements.

Kind regards,

Lai Kuan Lee

Academic Editor

PLOS ONE
---

## [Editor Report · Acceptance letter]

6 Sep 2022

PONE-D-22-01856R1 

Using dynamic microsimulation to project cognitive function in the elderly population 

Dear Dr. Wei:

I'm pleased to inform you that your manuscript has been deemed suitable for publication in PLOS ONE. Congratulations! Your manuscript is now with our production department. 

Kind regards, 

on behalf of

Dr. Lai Kuan Lee 

Academic Editor

PLOS ONE